# Global validation of the Particulate Observing Scanning Polarimeter (POSP) Aerosol Optical Depth products over land

Zhe Ji<sup>1,2</sup>, Zhengqiang Li<sup>1,2,3\*</sup>, Gerrit de Leeuw<sup>1,4</sup>, Zihan Zhang<sup>1</sup>, Yan Ma<sup>1</sup>, Zheng Shi<sup>5</sup>, Cheng Fan<sup>1</sup>, Qian Yao<sup>1,2</sup>

- <sup>1</sup> State Key Laboratory of Remote Sensing and Digital Earth & Key Laboratory of Satellite Remote Sensing of Ministry of Ecology and Environment, Aerospace Information Research Institute, Chinese Academy of Sciences, Beijing 100101, China <sup>2</sup> University of Chinese Academy of Sciences, Beijing 100049, China
  - <sup>3</sup> State Key Laboratory of Spatial Datum, College of Remote Sensing and Geoinformatics Engineering, Faculty of Geographical Science and Engineering, Henan University, Zhengzhou, 450046, China
- <sup>4</sup> Royal Netherlands Meteorological Institute (KNMI), R & D Satellite Observations, 3730 AE De Bilt, The Netherlands The Administrative Center for China's Agenda 21, Beijing 100098, China.

Correspondence to: Zhengqiang Li (lizq@radi.ac.cn)

Abstract. A global AOD retrieval product from the Particulate Observing Scanning Polarimeter (POSP) has been proposed. We have validated the AOD from the early stages of on-orbit operation and achieved high accuracy, but we lack an understanding of the retrieval accuracy over longer time scales from a systematic validation and analysis of POSP-retrieved AOD. The objectives of the current study are: 1) To ensure the reliability of POSP AOD products and explore the potential factors influencing their performance; 2) To provide a valuable reference for the enhancement of these products in future developments. To achieve these objectives, POSP AOD products have been validated using Aerosol Robotic Network (AERONET) measurements (over 276 sites) as reference. The results from 19314 collocations show a high accuracy, with correlation coefficients (R) of 0.914, a root mean square error (RMSE) of 0.085, and the fraction within the expected error (EE) of 78.5%. In addition, the validation at individual sites indicates that the performance of POSP products is better than that of MODIS (Deep Blue and Dark Target) AOD. Further error analysis indicates that the accuracy of POSP AOD exhibits a clear seasonal variation, being lower in the autumn and winter than in the spring and summer. Additionally, the uncertainty in AOD increases as NDVI decreases. Globally, the spatial variability of the quarterly averaged AOD has been analysed. The results show that the validation metrics of POSP and MODIS AOD are comparable. However, over North Africa and the Arabian Peninsula, POSP AOD is in better agreement with MAIAC AOD, while over other regions, it is in better agreement with DB/DT AOD.

#### 1 Introduction

Aerosols consist of particulate matter (referred to as particles) suspended in the atmosphere. Aerosols have increasingly attracted attention because of the air pollution caused by the rise in global industrial activity in recent years (Wei et al., 2023). Due to the large variation of sources, atmospheric aerosols have a wide variety of effects (de Leeuw et al., 2011). Aerosols can

directly and indirectly affect the radiative forcing of the Earth's climate (Stocker et al., 2013). Aerosol particles scatter solar radiation, thus reducing the amount of radiation that reaches the Earth's surface, thereby causing a cooling effect. In contrast, absorbing aerosols can absorb solar radiation, leading to a local warming effect (Guo et al., 2016). Aerosol particles can also act as cloud condensation nuclei (CCN), which in high relative humidity conditions can be activated and grow into cloud droplets. By influencing the CCN, aerosol particles can indirectly alter the microphysical properties of clouds (Myhre et al., 2007; Rosenfeld et al., 2014). Both effects depend on the size of the aerosol particles and on their composition. However, knowledge of the effects of atmospheric aerosols on climate is limited (Altieri et al., 2025). As a consequence, the Intergovernmental Panel on Climate Change (IPCC) considers aerosols to be one of the largest sources of uncertainty in global warming (Lee et al., 2023). Furthermore, aerosols are also harmful to human health (He and Huang, 2018; Li et al., 2017b). At the same time, high concentrations of aerosols significantly reduce near-surface visibility, crop production, etc (Hidy, 2019). Traditionally, the study of aerosol properties has mainly relied on ground-based observations. Through long-term investments and developments by various countries and their research institutions, a large number of ground-based observation sites have been established, providing optical and microphysical aerosol properties in key research areas (Dubovik et al., 2002; Levy et al., 2007). Examples include NASA's AErosol RObotic NETwork (AERONET) (Holben et al., 1998), Europe's PHOtométrie pour le Traitement Opérationnel de Normalisation Satellitaire (PHOTONS) (Goloub et al., 2008), China's Sun-sky radiometer Observation NETwork (SONET) (Li et al., 2018b), the China Aerosol Remote Sensing NETwork (CARSNET) (Che et al., 2015), etc. In addition to these ground-based remote sensing activities, a wide variety of ground-based in situ networks have been established on global (Moreno, 2023; Yuba et al., 2023), continental (Tørseth et al., 2012), and national (Bai et al., 2020; Beig et al., 2021; Cobourn, 2007; Hoff et al., 2006; Wu et al., 2018) scales. However, ground-based observations cannot provide data on global or regional scales and lack coverage in difficult-to-access areas. This gap can be filled by satellite observations, providing characterization of aerosols on regional to global scales, but with lower accuracy. From the perspective of sustainable development, the Chinese government has actively responded to the call for global environmental governance by launching numerous satellites for atmospheric environment monitoring (Chen et al., 2021; Wang et al., 2021; Xian et al., 2021; Zhao et al., 2017). Among these, the Particulate Observation Scanning Polarimeter (POSP), mounted on the Gaofen-5(02) satellite, was successfully launched in July 2021. POSP is a single-view multispectral polarimeter with a nadir resolution of 6.4 km.

Aerosol Optical Depth (AOD) is the primary parameter used to assess the atmospheric aerosol content by remote sensing methods. Observations from POSP provide valuable information for the retrieval of aerosol properties (Li et al., 2022). An aerosol retrieval algorithm for application to POSP data was initially developed by Shi et al. (2023), who developed a method for the reconstruction of the land surface reflectance for the use in aerosol retrieval. Then, Ji et al. (2025) proposed a more accurate aerosol algorithm that takes into account the directional properties of the surface. By exploring the empirical relationships between adjacent blue bands, the inversion of AOD has been realized with this new algorithm by jointly using multiple blue bands. An optimization algorithm has also been used to incorporate boundary constraints, which simultaneously

65 accounts for errors in the surface constraint and the satellite observations. Thus, POSP AOD products are successfully retrieved with high accuracy.

We have validated the retrievals from the early stages of on-orbit operation (November 2021 to April 2022) and achieved high accuracy, but lack an understanding of the retrieval accuracy over longer time scales. To further validate the accuracy of the POSP AOD product and to identify the direction of further improvement, we validated the AOD product for a longer time series from December 2021 to November 2022. Comparisons of POSP AOD with AERONET AOD show a high degree of consistency, and its accuracy surpasses that of MODIS AOD for the same time period (Ji et al., 2025).

This study is dedicated to a comprehensive evaluation of POSP AOD products, delving into the exploration of potential factors influencing their performance. The ultimate goal is to provide a valuable reference for the further improvement of these products in future developments. The definitions of the statistical metrics used are presented in Section 2. The matchup strategies and data preprocessing are presented in Section 3. The validation of the results is presented in Section 4. Section 5 discusses the error analysis of POSP AOD in different seasons and over different land surface types, characterized by land cover (LC), the comparison of POSP and MODIS AOD, and the time-series analysis over some of the most polluted cities. Conclusions are presented in Section 6.

## 2 Materials

## 2.1 POSP AOD products

The POSP was launched on board the GF-5(02) satellite in July 2021. It has a field of view of  $\pm 50^{\circ}$  with a swath width of ~1850 km, and provides global observations in nine spectral bands spanning wavelengths from 380 to 2250 nm (Lei et al., 2023). The local time of the descending node for GF-5(02) is 10:30 a.m. The POSP is equipped with a comprehensive onboard calibration system (the radiometric calibration accuracy is within 5%, and the polarimetric calibration accuracy is within 0.005). Ji et al. (2025) developed an enhanced AOD retrieval algorithm using POSP data. Due to the limited number of observations, POSP faces an ill-posed inversion problem when the directional characteristics of the surface are taken into account. To reduce the discrepancy between the number of observations and the number of retrieval parameters, the following changes have been made to the algorithm presented in Ji et al. (2025). For aerosol parameters, the global aerosol distribution from the MODIS Dark Target algorithm has been used, but aerosol models over northern India and central Africa have been updated to achieve more accurate retrievals. For surface parameters, the bidirectional reflectance distribution function (BRDF) from MODIS (MCD43) was used to account for the directional reflectance characteristics of the surface during the inversion (Schaaf et al., 2002). The MODIS BRDF comprises an isotropic kernel (reflectance from Lambertian surface), a volumetric kernel (reflectance from multiple scattering within vegetation canopies), and a geometric-optical kernel (reflectance from object shadowing). To eliminate the differences in spectral response between POSP and MODIS, spectral reconstruction was performed using the Singular Value Decomposition (SVD) technique. The algorithm only retrieves the isotropic kernel to reduce the number of parameters to be inverted. Therefore, after spectral reconstruction, monthly averaged Ross-Thick and

Li-Sparse kernel parameters were applied to account for the surface directional characteristics. Finally, the new aerosol models and surface directional characteristics were incorporated into the algorithm developed by Ji et al. (2025), and AOD was successfully retrieved. Ji et al. (2025) also presented the preliminary validation (from November 2021 to April 2022), the results show that the AOD retrievals have high accuracy.

## 2.2 MODIS products

The MODIS AOD product has been operationally available for many years, with several updates to the most recent C6.1 released in 2017 (Sayer et al., 2017). MODIS AOD products have demonstrated stability through extensive validation (Levy et al., 2013; Sayer et al., 2013). In this study, we selected the MODIS/Terra C6.1 aerosol products (Level 2.0) with a spatial resolution of 10 km from the Deep Blue (DB) and Dark Target (DT) algorithms for comparison. Furthermore, the MODIS AOD products (MOD04) have quality flags (QA), with QA=3 representing the highest quality. In the cross-validation with MODIS, we only used data with QA=3.

To quantitatively assess the accuracy of the POSP AOD algorithm over different surface types, the land cover (LC) product MCD12Q1 in 2022 was used in this study (Sulla-Menashe and Friedl, 2018). The International Geosphere-Biosphere Programme (IGBP) classification scheme was applied, which has been widely used in climate and environmental studies as a global classification standard for describing land cover types. The global IGBP classification results for 2022 are depicted in Figure 1. To match the POSP pixel size, the MCD12Q1 results were resampled to  $0.005^{\circ} \times 0.005^{\circ}$ . Then the IGBP type for each AERONET station was obtained within a  $40 \times 40$  pixels window centred on the AERONET station. The LC type with the highest occurrence in this window was selected as representative for each AERONET station. Because only a few sites have homogeneous LC types in their surrounding areas; therefore, we use the LC type with the highest occurrence to represent the LC type within that spatial window. MODIS products are available from <a href="https://ladsweb.modaps.eosdis.nasa.gov/">https://ladsweb.modaps.eosdis.nasa.gov/</a>.

Figure 1: The distribution of selected AERONET sites globally. The red points represent inland sites. The background is the MCD12C1 land cover classification product (IGBP) in 2022. Light Blue: Water Bodies (WB); Dark Olive Green: Evergreen Needleleaf Forests (ENF); Forest Green: Evergreen Broadleaf Forests (EBF); Dark Green: Deciduous Needleleaf Forests (DNF);

Bright Green: Deciduous Broadleaf Forests (DBF); Light Forest Green: Mixed Forests (MF); Brick Red: Closed Shrubland (CS); Tan: Open Shrublands (OS); Light Brown: Woody Savannas (WS); Light Orange: Savannas (Sa); Oliver Green: Grasslands (Gr); Deep Blue: Permanent Wetlands (PW); Mustard Yellow: Croplands (Cr); Bright Red: Urban and Built up Lands (UB); Dark Olive: Cropland Natural Vegetation Mosaics (CNVM); Light Grey: Permanent Snow and Ice (SI); Pale Yellow: Barren (Ba).

## 125 **2.3 AERONET data**

AERONET provides aerosol products with low uncertainties: 0.01 in the VIS range and 0.02 in the UV range (Eck et al., 1999; Giles et al., 2019). AERONET AOD is extensively used as a reference for satellite validation (Che et al., 2016; Chu et al., 2002; Levy et al., 2010; Sayer et al., 2013; Xie et al., 2019). AERONET V3 provides AOD datasets at three quality levels: Level 1.0 following pre-screening, Level 1.5 after cloud identification and instrument anomaly monitoring, and Level 2.0 after cloud identification, instrument anomaly monitoring, and quality control screening (<a href="https://aeronet.gsfc.nasa.gov/">https://aeronet.gsfc.nasa.gov/</a>). In this study, Level 1.5 data is chosen as the ground-based validation data to minimize validation errors.

Since POSP AOD products are produced at a wavelength of 550 nm, which is not available from AERONET, AERONET AOD data were interpolated to 550 nm using observation at 440 and 675 nm and the Ångström Exponent (AE) (Ångström, 1929).

35 
$$\begin{cases} \tau_{\lambda} = \beta \lambda^{-\alpha} \\ \alpha = -\left(\ln\left(\tau_{\lambda_{1}}/\tau_{\lambda_{2}}\right)\right)/(\lambda_{1}/\lambda_{2}) \end{cases}$$
 (1)

where  $\lambda$  is the specified wavelength in nm,  $\tau_{\lambda}$  is the AOD at wavelength  $\lambda$ , and  $\alpha$  is the AE. AE is calculated using the AOD at wavelengths  $\lambda_1$  (440 nm) and  $\lambda_2$  (675 nm).

The POSP AOD algorithm is only applicable over land and cannot provide aerosol data over the ocean and in coastal regions.

In this study, stations within 20 km of the coastline are defined as coastal stations and are excluded from the validation to ensure the reliability of the results. As a result, 276 sites remain for validation.

## 3 Methods



# 3.1 Matchup strategy

For the collocation of satellite and AERONET AOD data, various spatial and temporal matchup strategies have been proposed (Chu et al., 2002; Ichoku et al., 2002; Sayer et al., 2013; Virtanen et al., 2018). In this study, considering the 6.4 km spatial resolution of the POSP, the following strategies to match POSP and AERONET AOD data have been devised to ensure reliable AOD validation results while accounting for spatial consistency: satellite data are averaged over a window of 3 × 3 pixels centred on the AERONET site, and ground-based observations are averaged over 30 minutes before and after the time of the satellite overpass. To mitigate the uncertainty associated with averaging data, a minimum of two or more ground-based observations are required in the temporal matchup window, and the spatial-temporal matchup window must encompass more than three valid satellite pixels (Chu et al., 2002). The POSP and MODIS AOD matchup data pairs at the same AERONET site on the same day were used for comparison. To investigate the influence of land cover (LC) on the POSP AOD retrieval,

the validation was repeated for sub-sets of POSP data over different IGBP types. The number of matchups over forested areas (evergreen broadleaf forest, evergreen needleleaf forest, deciduous broadleaf forest, and mixed forest) was too small to achieve statistical significance, and therefore, they were merged into the "Forest" category. Likewise, for shrublands with low vegetation (woody savannas, grassland, and savanna), the data were merged into the "Grassland" category.

## 3.2 Statistical metrics



To quantitatively assess the accuracy of the retrieval results and the applicability of the retrieval algorithms over different surface types, statistical metrics were calculated for the validation. These metrics include the Pearson correlation coefficient (R), which reflects the degree of agreement between the satellite retrieval results and the ground-based reference data.

$$R = \frac{\sum_{i=1}^{n} (AOD_{AERONET,i} - \overline{AOD_{AERONET}})(AOD_{Satellite,i} - \overline{AOD}_{Satellite})}{\sqrt{\sum_{i=1}^{n} (AOD_{AERONET,i} - \overline{AOD_{AERONET}})^{2} \sum_{i=1}^{n} (AOD_{Satellite,i} - \overline{AOD}_{Satellite,i})^{2}}},$$
(2)

where  $AOD_{AERONET,i}$  denotes the AERONET reference data,  $AOD_{Satellite,i}$  denotes the satellite retrieval data, n denotes the number of collocations, and  $\overline{AOD_{AERONET}}$  and  $\overline{AOD_{Satellite}}$  denote the averages of ground and satellite results, respectively. R<sup>2</sup> (R-squared), also known as the coefficient of determination, represents the difference between the satellite retrievals and the ground-based reference data. Its value ranges from 0 to 1, with values closer to 1 or 0 indicating the difference between them is small or large, respectively.

$$R^{2} = 1 - \frac{\sum_{i=1}^{n} \left(\frac{AOD_{AERONET,i} - AOD_{Satellite}}{\sum_{i=1}^{n} \left(\frac{AOD_{AERONET,i} - \overline{AOD_{AERONET}}}{AOD_{AERONET,i}}\right)^{2}},$$
(3)

Root Mean Square Error (RMSE) presents the uncertainty in the results of the satellite retrievals with respect to the ground-based reference data.

$$RMSE = \sqrt{\frac{1}{n} \sum_{i=1}^{n} \left( AOD_{Satellite,i} - AOD_{AERONET,i} \right)^{2}},$$
(4)

Mean Absolute Error (MAE) indicates the overall estimation of the accuracy of the retrieval results.

$$MAE = \frac{1}{n} \sum_{i=1}^{n} |AOD_{Satellite,i} - AOD_{AERONET,i}|,$$
(5)

Bias is a measure of underestimation or overestimation with respect to the reference data.

$$Bias = \frac{1}{n} \sum_{i=1}^{n} (AOD_{Satellite,i} - AOD_{AERONET,i}), \tag{6}$$

Furthermore, the accuracy of the AOD retrievals is assessed in this study using a combination of absolute and relative errors, referred to as the expected error (EE). EE represents the theoretically expected standard deviation of the AOD product and thus indicates the boundaries within which 67% of the matchup data pairs should fall (Xie et al., 2019). In this study, we adopt the EE, which applies to the MODIS Collection 6.1 (C6.1) AOD products, enabling a comparison of accuracy with MODIS AOD products using the same criteria (Levy et al., 2010, 2013):

$$EE = \pm (0.05 + 0.15 * AOD), \tag{7}$$

We also implemented the criteria proposed by the Global Climate Observing System (GCOS), which have been adopted in the Aerosol cci study (Popp et al., 2016; Secretariat, 2006).

$$GCOS = maximum(0.03, 0.1 \times AOD), \tag{8}$$

# 3.3 Data preprocessing




As an optical sensor, POSP observations are inherently susceptible to cloud interference. To mitigate cloud contamination, it is essential to filter out cloud-affected pixels before retrieval. Given the single-angle observation method of POSP, this study adopts cloud detection strategies from MODIS, which have been extensively validated (Frey et al., 2008). Specifically, two methods are employed: the apparent reflectance threshold method and the apparent reflectance spatial variation detection method (Martins et al., 2002). The former effectively identifies optically thick clouds with high reflectance or substantial water vapor content, while the latter is particularly useful for detecting cloud edges, shadows, thin clouds, and dispersed cloud formations.

The land surface exhibits low reflectance in the blue band, whereas clouds have high reflectance. Therefore, a pixel is identified as a cloud when its reflectance at the 443 nm band exceeds a certain threshold. The 1380 nm band lies within a strong water vapor absorption region, where the reflectances from land surfaces and low clouds are generally low. As a result, only high clouds, mostly above the heights where atmospheric water vapor is located, are visible in this band. Pixels with high reflectance at 1380 nm are therefore typically classified as high clouds. Furthermore, cloud edges typically exhibit high spatial variability due to mixed pixels and partial cloud coverage. The spatial variation characteristics of the 443 nm and 1380 nm bands can effectively identify cloud-edge pixels. The combination of their spatial differences helps reduce misclassification at cloud boundaries and improves the accuracy of cloud detection.

Surface conditions such as snow and water also affect the inversion. Since the retrieval algorithm is explicitly designed for clear-sky over non-ice land surfaces, pixels over water, ice, and snow must be excluded. The detection of water and snow 200 pixels is achieved using the Normalized Difference Water Index (NDWI) and the Normalized Difference Snow/Ice Index (NDSI), respectively, with specific identification thresholds presented in Table 1.

$$NDWI = \frac{\rho_{670} - \rho_{865}}{\rho_{670} + \rho_{865}} \tag{9}$$

$$NDWI = \frac{\rho_{670} - \rho_{865}}{\rho_{670} + \rho_{865}}$$

$$NDSI = \frac{\rho_{670} - \rho_{2250}}{\rho_{670} + \rho_{2250}}$$
(10)

While the aforementioned cloud detection strategy provides a foundation for minimizing cloud contamination, potential for 205 further improvement remains. Given the relatively coarse spatial resolution of POSP (6.4 km) and its limited spectral coverage, certain pixels that contain residual clouds may remain undetected. The simulation analysis by Kassianov and Ovtchinnikov (2008) pointed out that multiple scattering of clouds can lead to overestimated AOD retrievals when the residual clouds are not fully screened. Sogacheva et al. (2017) further removed the cloud-contaminated pixels using a cloud post-processing scheme. To enhance cloud-mask accuracy, a dedicated cloud detection algorithm for POSP is still needed. We aim to further enhance the cloud detection algorithm in future work.

Table 1 Summary of screening thresholds.

| Items                                              | Purpose  |  |  |
|----------------------------------------------------|----------|--|--|
| $ \rho_{443} < 0.02 \text{ or } \rho_{443} > 0.4 $ | Cloud    |  |  |
| $\sigma_{443} > 0.038$                             | Cloud    |  |  |
| $ ho_{1380} > 0.02$ and Height $< 1500$            | Cloud    |  |  |
| $\sigma_{1380} > 0.005$                            | Cloud    |  |  |
| NDWI > 0                                           | Water    |  |  |
| NDSI > 0.4                                         | Snow/Ice |  |  |

# 4 Results



# 4.1 Overall validation

Figure 2 (A) shows the validation of POSP AOD in 2022 using AERONET AOD as reference, with R of 0.914, R<sup>2</sup> of 0.825, RMSE of 0.086, MAE of 0.054, and the fraction within EE is 78.45%.

The probability density functions of differences (POSP-AERONET) are presented in Figure 2 (B). The results show that the POSP algorithm underestimates the AOD as aerosol loading increases. For low AOD (AOD < 0.2), POSP's bias is 0.01. For moderate AOD ( $0.2 \le AOD \le 0.7$ ), POSP's bias increases to -0.03, and for high AOD (AOD > 0.7), POSP's bias further increases to -0.04. These biases may be attributed to the increasing aerosol model error. As AOD increases, the impact of discrepancies between the assumed aerosol model and the actual aerosol model is amplified, leading to an increase in retrieval uncertainty (Hou et al., 2018; Li et al., 2018a). Box plots of differences between POSP and AERONET AOD against AERONET AOD in Figure 2 (C) show how the AOD bias is distributed across different AOD intervals. With the increase in aerosol loading, the AOD bias overall increases (more negative) but exhibits an anomaly at high AOD, spiking from negative to positive values. Except for the aerosol model error, the possible reason for this anomaly may be that the frequency of high AOD decreases with increasing AOD, and the smaller statistical sample introduces a greater uncertainty. Further research is needed on this phenomenon when results are available for a longer period of time.

Figure 2: (A) Scatter density plot of POSP AOD versus AERONET AOD, where N—number of collocated data pairs, R—Pearson correlation coefficient, RMSE—root mean square error, MRE—mean relative error, and EE—data fraction within EE. The black dotted line represented the one-to-one line. The red line represents the linear regression fit, and the black dashed lines are EE lines. The magenta points indicate the mean values of the satellite AOD binned in AERONET AOD intervals varying from 0.01 for small AOD to 0.25 for the larger AOD up to 2.0. The magenta lines are the ±2σ of the retrieved in each AERONET bin. (B) Probability density plots of differences (POSP-AERONET). The black, blue, green, and red solid lines indicate different AOD conditions: all AOD, AOD < 0.2, 0.2 ≤ AOD ≤ 0.7, and AOD > 0.7, respectively. (C) Box plots of the differences between POSP AOD and AERONET AOD. The blue dots and error bars represent the median, 25th percentiles, and 75th percentiles of the AOD bias.

# 4.2 Validation of POSP AOD in different seasons




Atmospheric and aerosol conditions vary with the seasons (Bergametti et al., 1989; Rabha and Saikia, 2020). Thus, the difficulties of AOD retrieval also change between seasons (Che et al., 2016; Fan et al., 2023; He et al., 2016). Figure 3 shows that POSP AOD is more accurate in SON (September, October, November) and DJF (December, January, February) than in the MAM (March, April, May) and JJA (June, July, August). For comparison, similar plots for the MODIS DB/DT AOD product are presented in Figures S1 and S2, showing that the MODIS AOD accuracy in SON and DJF is significantly better than in MAM and JJA. The validation of POSP AOD during different seasons shows that POSP AOD has the highest accuracy in DJF, with the fraction within EE being 82.13%. SON is second best, with the fraction within EE of 79.85%. JJA has the lowest accuracy, with the fraction within EE being 75.98%. The POSP AOD has a very low bias during all seasons. The higher accuracy of both POSP AOD and MODIS AOD during DJF and SON compared to MAM and JJA may be attributed to the fact that most of the AERONET sites used for validation are located in the Northern Hemisphere (see Figure 1). DJF and SON correspond to winter in the Northern Hemisphere, a period when surface changes are slower, and sudden pollution events are less frequent. As a result, the empirical constraints used in the retrieval process are more effective compared to those applied during the summer. The number of collocations is smallest in DJF (N=3834) and highest in JJA (N=5454). A possible reason for the difference between winter (DJF) and summer (JJA) is that snow and ice cover reduce the number of successful retrievals in the winter in the Northern Hemisphere.

Figure 3: Upper panels show scatter density plots of POSP AOD versus AERONET AOD for different seasons: (A) DJF, (B) MAM, (C) JJA, and (D) SON. The middle panels show the probability density functions of differences (POSP-AERONET). Lower panels show box plots of the difference between POSP AOD and AERONET AOD. See the Figure 3 caption for further explanation of the various features plotted.

## 4.3 Validation of POSP AOD over different surface types



Kaufman et al. (1997) pointed out that a small error of 0.01 in surface reflectance can lead to a 0.1 uncertainty in the retrieved AOD. To evaluate the influence of surface type, the validation results of POSP AOD over four different groups of LC types, city, cropland, grassland, and forest, are plotted in Figure 4. The results show that the validation metrics vary with LC type. Over forest, the RMSE is lowest and the fraction within EE (79.92%) is highest, but the AOD range is limited and high AOD cases are lacking, while also R is lowest (0.85). Over Cropland, Grassland, and Forest, the statistical metrics are similar, while City has the lowest accuracy. In addition, the AOD over Cropland, Grassland, and Forest is nearly unbiased (less than 0.01), while the AOD over City shows a positive bias. The reasons for the lower accuracy over City than other surface types will be discussed in Section 5.1.3.

Figure 4: POSP AOD validation results over four different land cover types: (a) city, (b) cropland, (c) grassland, and (d) forest. See the Figure 3 caption for further explanation of the various features plotted.

## 4.4 Site-specific validation metrics



To evaluate the reliability of the POSP AOD product over different regions, the values of four validation metrics (R, RMSE, bias, and GCOS) are plotted on global maps in Figure 5 (similar maps for MODIS DB/DT AOD are presented in Figures S3 and S4). For most sites, the accuracy of the POSP AOD is high. As indicated in Table 2, 24%, 50%, and 78% of the sites have RMSEs less than 0.05, 0.07, and 0.1, respectively. Additionally, 22%, 57%, and 86% of the sites have GCOS fractions greater than 60%, 45%, and 30%, respectively. Figure 5 (A) shows that in North America and Europe, R is slightly lower than in other regions. However, Figure 5 (B) shows that in North America and Europe, the RMSE is closer to zero than in other regions. Figure 5 (D) indicates that in these regions, the GCOS fraction is much higher than in other parts of the world. Additionally, Figure 5 (C) shows that the sites in North America and Europe have a slightly positive bias. This is because the AOD in these regions is low, and the lower spatial resolution of the POSP results may be affected by residual clouds, leading to overestimation. The sites in Africa and India with deep colour show a larger positive bias than in North America and Europe. This is attributed to the higher aerosol loading in these regions, which results in increased retrieval uncertainty.

On the other hand, in heavily polluted regions such as northern India, central and western Africa, and central South America, POSP AOD shows high consistency with AERONET AOD, although the GCOS fraction is lower. This is because a fixed aerosol model is used to improve the stability of the inversion, which, however, may not accurately represent the actual aerosol

types. Such discrepancies introduce greater uncertainties in the retrieval as the aerosol loading increases. Thus, using a fixed aerosol model inevitably affects the retrieval accuracy (Levy et al., 2013). This is one of the inherent challenges of aerosol retrieval using single-angle observations, and we aim to address this issue in future algorithm improvements.

Figure 5: Global maps showing site-specific metrics for the validation of POSP AOD using AERONET AOD as reference: (A) R, (B) RMSE, (C) bias, and (D) the fraction within GCOS.

Table 2: Percentages of exceedance of discrete values of POSP AOD validation metrics.

|      | > 0.7     | > 0.6     | > 0.5     |  |  |
|------|-----------|-----------|-----------|--|--|
| R    |           |           |           |  |  |
|      | 123 (45%) | 173 (64%) | 216 (79%) |  |  |
| RMSE | < 0.05    | < 0.07    | < 0.1     |  |  |
|      | 65 (24%)  | 137(50%)  | 213 (78%) |  |  |
| Bias | < 0.04    | < 0.02    | < 0.01    |  |  |
|      | 226(83%)  | 182(77%)  | 109(40%)  |  |  |
| GCOS | >60%      | >45%      | >30%      |  |  |
|      | 61(22%)   | 154(57%)  | 266(86%)  |  |  |

# 5 Discussion

# 5.1 Error Analysis

# 5.1.1 Error analysis of AOD bias in different seasons

To further assess the impact of different factors on the accuracy of POSP AOD retrievals, Figure 6 shows the variation of AOD bias with the AE evaluated from AOD at 670 nm and 865 nm (AE<sub>670-865</sub>), scattering angle, and Normalized Difference

Vegetation Index (NDVI), for each of the four seasons. For AE<sub>670-865</sub> varying between 0.25 and 1.55, the values of the mean bias are all similar, in any of the seasons, indicating that the algorithm performs well regardless of the particle sizes. However, the 25 and 75 percentiles show that substantial variations occur and that these variations are largest for both the lowest and highest AE values shown. The bias decreases somewhat with increasing NDVI, with the largest decrease for the larger NDVI values. Furthermore, the bias uncertainty, represented by the length of the error bars. decreases as NDVI increases. The latter indicates that the POSP AODs are more accurate over densely vegetated areas than over low-vegetated areas. The AOD bias varies somewhat with the scattering angle and increases at the largest scattering angles. The bias uncertainty increases with increasing scattering angle, for all seasons except in DJF and SON, when it decreases at scattering angles of 170° and 180°. The data in Fig. 6 show that the variations in bias with AE<sub>670-865</sub>, scattering angle, and NDVI are not significantly influenced by seasonal changes.

Figure 6: Box and whisker plots of AOD bias as a function of (1) AERONET AE<sub>440-870</sub>, (2) POSP NDVI, (3) scattering angle, in (A) DJF, (B) MAM, (C) JJA, and (D) SON. The black dots and error bars represent the median, 25th percentiles, and 75th percentiles of the AOD bias.

## 5.1.2 Error analysis of AOD bias over different land cover types



Figure 7 shows the AOD bias over different LC types using the data from the full study period (The conclusion from section 5.1.1 indicates that the seasonal influence on the retrieval bias is negligible). The patterns for AE<sub>670-865</sub> and NDVI are similar for all four types of LC. In contrast, the effect of the scattering angle is much larger over the city than over the other three areas, especially for scattering angles larger than 135°, where the AOD is substantially overestimated. This is because the hotspot effect becomes more pronounced with increasing scattering angle (Jiao et al., 2016), where the hotspot effect refers to an anisotropic scattering phenomenon characterized by a notable increase in observed reflectance when the solar illumination and sensor viewing directions coincide (Bréon et al., 2002). As a consequence, an error in the surface reflectance results in an increased uncertainty in the AOD retrievals. It is important to note that the hotspot effect was not considered when estimating surface reflectance. This effect is more pronounced over cities than over other surfaces because a city has more complex surfaces and varied pollution components (Bilal et al., 2022; Wong et al., 2011). The impact of neglecting surface directional reflectance characteristics over urban areas will be discussed in Section 5.1.3.

Figure 7: Box and whisker plots of the POSP AOD bias versus AERONET AE<sub>670-865</sub>, POSP NDVI, and scattering angle for four different land cover types: (A) city, (B) cropland, (C) grassland, and (D) forest.

## 5.1.3 The impact of neglecting surface directional reflectance characteristics over city areas on the retrieval






To further explore the impact of urban surface reflectance anisotropy on aerosol retrieval, synthetic experiments have been made. Following the detailed description of the spectral reconstruction of BRDF kernel coefficients in Section 2.1, here the BRDF kernel coefficients are filtered further for urban LC using the global IGBP classification product MCD12C1 (The MCD12C1 product was resampled to match the spatial resolution of the BRDF results). The number of BRDF kernel coefficients obtained over urban LC is quite large, making it impractical to compute TOA reflectance under different observation geometries and aerosol conditions for each individual case. To simplify the computation while retaining the representativeness of BRDF kernel coefficients over urban areas, we applied the K-means clustering method to extract BRDF kernel coefficients representative of urban areas in 2022 (tests showed that seven clusters are sufficient to represent the urban BRDF kernel coefficients). The results are presented in Table 3.

To evaluate the effect of ignoring surface directional characteristics over urban areas on the retrieved aerosol properties, the non-Lambertian radiative transfer model (RTM) and Lambertian RTM are used for creating synthetic TOA reflectances ( $\rho'$ ) and AOD retrieval results, respectively. In the retrieval process, the Lambertian RTM is used to calculate the TOA reflectance ( $\rho^*$ ). The AOD corresponding to the best match between  $\rho^*$  and  $\rho'$  is taken as the retrieval result. By comparing the retrieval bias, the effect of ignoring surface anisotropy on AOD retrieval over urban areas was assessed.

The calculation of TOA reflectance requires consideration of three aspects: For the aerosol properties, to simulate aerosol conditions over urban areas, we used a mixture of continental and polluted aerosol types in equal proportions (Omar et al., 2009). And AOD was set to range from 0 to 1.5. Furthermore, in order to reduce the influence of errors introduced by aerosol model uncertainty, the aerosol model used in the retrieval is the same as that used in the creation of the synthetic dataset. For the surface reflectance, we used the seven representative BRDF kernel coefficients derived from the above clustering process. When calculating TOA reflectance using the non-Lambertian RTM, all three BRDF kernel coefficients from Table 3 are used to estimate the surface reflectance. In contrast, when using the Lambertian RTM, only the isotropic kernel coefficient from Table 3 is used as the surface reflectance. For the observation geometries, the solar zenith angle, viewing zenith angle, and relative azimuth angle were set to range from 10° to 70°, 0° to 60°, and 0° to 360°, respectively. The observation geometry and AOD were both randomly sampled following a Gaussian distribution. To account for real-world observational conditions, we introduced random errors to the simulated reflectances consistent with calibration accuracy.

Figures 8 and 9 show polar diagrams of the surface reflectances, calculated using the MODIS BRDF model (Eqs. S1 to S9, with the solar zenith angle set at 30°), for each of the seven LC for urban types, as well as the scattering angle. These calculations were made for wavelengths used in the POSP retrieval algorithm, i.e., at 443 nm (Fig. 8) and 490 nm (Fig. 9). The simulations show that the surface reflectance increases significantly for viewing zenith angles larger than approximately 75°. In the retrieval algorithm, this issue is avoided by restricting the viewing zenith angles to less than 60°. Furthermore, surface reflectance increases substantially when the viewing zenith angle approaches the solar zenith angle, corresponding to the

maximum scattering angle. This explains the high uncertainty over urban areas at large scattering angles discussed in Section

360 5.1.2.

Table 3 BRDF kernel coefficients statistics over urban areas for different types.

| Kernal name       | Band<br>(nm) | Type 1 | Type 2 | Type 3 | Type 4 | Type 5 | Type 6 | Type 7 |
|-------------------|--------------|--------|--------|--------|--------|--------|--------|--------|
| Isotropic         | 443          | 0.068  | 0.044  | 0.053  | 0.104  | 0.076  | 0.049  | 0.127  |
|                   | 490          | 0.083  | 0.055  | 0.065  | 0.123  | 0.091  | 0.060  | 0.150  |
| volumetric        | 443          | 0.017  | 0.021  | 0.017  | 0.028  | 0.041  | 0.036  | 0.058  |
|                   | 490          | 0.021  | 0.024  | 0.022  | 0.033  | 0.048  | 0.042  | 0.067  |
| geometric-optical | 443          | 0.016  | 0.010  | 0.011  | 0.023  | 0.014  | 0.007  | 0.020  |
|                   | 490          | 0.020  | 0.013  | 0.013  | 0.027  | 0.017  | 0.009  | 0.023  |

Figure 8: Polar diagrams of the BRDF distribution for the 7 types of clustered results. (A)-(F) The result of surface reflectance at 443 nm, and (G) The scattering angle plot. In this polar plot, the radius denotes a change in viewing zenith angle from 0° to 90°, and the polar angle represents a change in relative azimuth angle from 0° to 360°. The simulations are performed for a solar zenith angle of 30°. The colors in (A)-(F) and (G) represent surface reflectance and scattering angle magnitude, respectively.

Figure 9: As Figure 8, but for a wavelength of 490 nm.

Figures 10 and S7 show the AOD bias as a function of scattering angle for 7 different surface types to illustrate how retrieval errors caused by neglecting surface anisotropy vary with scattering angle and aerosol loading, respectively. Because of the overestimation of the simulated reflectance using the Lambertian forward radiative transfer model, the retrieved AOD is underestimated. For types 4 and 7, which have the highest reflectance, the AOD underestimation is most pronounced, confirming that the higher the surface reflectance, the greater the impact of ignoring surface anisotropy on retrieval accuracy. For types 2 and 3, which have the lowest reflectance surfaces, the retrieval error caused by neglecting surface anisotropy is nearly constant across different aerosol loadings (Figure S7), but slightly increases as the scattering angle increases. Overall, as AOD increases, the impact of ignoring surface anisotropy on retrievals diminishes, and as surface reflectance increases. Therefore, for aerosol retrieval over urban areas, the effect of surface anisotropy on the retrieval result is non-negligible in regions with high surface reflectance.

Figure 10: AOD bias as a function of scattering angle for the 7 different types of urban LC clusters

# 5.2 Comparison of POSP and MODIS AOD

## 5.2.1 Overall validation

Figure 11 shows comparisons of the POSP/GF-5(02) and MODIS/Terra AOD versus AERONET data, where the MODIS AOD includes DB (11,010 collocations) and DT (9,211 collocations). The number of DB and DT collocations is different because DT does not provide retrieval results over bright surfaces (The validation for DB/DT is presented in Figures S1 and S2, respectively).

The comparison in Figure 11 shows that the POSP/GF-5(02) AOD has a higher accuracy than the MODIS/Terra DB AOD, with the fractions within EE of 82.5% and 77.3%, respectively. Likewise, POSP/GF-5(02) has higher accuracy than MODIS/Terra DT AOD, with the fractions within EE of 80.7% and 73.9%, respectively. The probability distribution functions in Figure 11 show that POSP and DB are nearly unbiased, while DT slightly overestimates AOD. These results show that the accuracy of POSP/GF-5(02) AOD is overall better than that of MODIS/Terra AOD for both DB and DT.

Figure 11:The upper panels show scatter density plots of satellite AOD versus AERONET reference data, where POSP AOD is plotted in red and MODIS AOD in blue; The lower panels show the probability distribution functions for the differences between satellite AOD and AERONET AOD. The left column represents the results for the DB algorithm (A) and the right column represents the DT algorithm (B).

## 5.2.2 Validation over different surface types

The comparison of POSP/GF-5(02) and MODIS/Terra AOD over four different land cover types (city, cropland, grassland, and forest) is presented in Figure 12 for DB and in Figure 13 for DT. The accuracy of POSP is higher than that of both DB and DT over cropland, grassland, and city. However, over the Forest, the accuracy of POSP/GF-5(02) AOD is lower than that of DB (Figure 12). Specifically, in terms of R<sup>2</sup>, RMSE, and Bias, the metrics are better for POSP AOD than for DB, but not for other accuracy metrics. The comparison of bias histograms over different land cover types indicates that the POSP AOD is nearly unbiased over all surface types, whereas DB shows a positive bias over city, and DT shows a positive bias over city, cropland, and forest.

Figure 12: Same as Figure 11, for AOD from POSP/GF-5(02) and MODIS/TERRA DB, but over different land cover types: (A) city, (B) cropland, (C) grassland, and (D) forest.

Figure 13: Same as Figure 12, but for DT.



# 5.2.3 Comparison of the spatial distributions of AOD from POSP and MODIS/Terra

Figure 14 shows the spatial distributions of seasonally averaged AOD from POSP/GF-5(02) and MODIS/Terra, for MAIAC, DB, and DT (top to bottom). Difference plots between the seasonally averaged POSP/GF-5(02) and MODIS/Terra AOD are presented in Figure S5. During DJF, AOD could not be retrieved over high-latitude regions due to the presence of snow and ice. In addition, the low solar zenith angle over high-latitude regions will also affect the inversion. The seasonal variation of

the spatial characteristics is similar to that of other satellite products (Chen et al., 2020; Fan et al., 2023). In 2022, AOD was higher during MMA and JJA than during DJF and SON. In addition, AOD is lower due to stable atmospheric conditions and reduced atmospheric vertical convective activity during winter (Liu et al., 2022; Zhao et al., 2018). Frequent biomass burning events contribute to elevated AOD in south-central Africa (Tummon et al., 2010). Furthermore, high AOD persists in eastern China and northern India due to active industrial production and biomass burning events (de Leeuw et al., 2018; Gupta et al., 2021).



POSP AOD is slightly lower than MODIS DB AOD over North Africa and the Arabian Peninsula, while it is much closer to MODIS MAIAC AOD in these regions. Overall, POSP AOD shows similar features as MAIAC AOD over North Africa and the Arabian Peninsula, while it is more consistent with DB AOD over other regions. Furthermore, compared to MODIS DB, the spatial differences between POSP AOD and MODIS DT are smaller. In general, the results indicate a high degree of agreement between POSP and MODIS AOD, with differences predominantly within the (-0.2, 0.2) range.

Figure 14: Maps of the seasonally averaged AOD derived from POSP, MODIS MAIAC, MODIS DB and MODIS DT, for the winter (DJF: December–January–February), spring (MAM: March–April-May,) summer (JJA: June–July–August), and autumn (SON: September–October–November).

## 5.3 Time-series analysis




Time series of POSP AOD are presented in Figure 15 for four polluted regions, together with MODIS/TERRA DB and AERONET AOD data, for the whole year 2022. Figure 15 (A) shows these time series over the Beijing CAMS site in Beijing in the North China Plain (NCP), which is a key region for aerosol research due to its unique economic and geographical characteristics (Deng et al., 2011; Li et al., 2017a; Liu et al., 2011). The Beijing CAMS site is located within Beijing. For POSP AOD, the RMSE is lower than for MODIS DB AOD, and POSP has more valid retrievals. Severe pollution events (AOD > 1.5) were recorded in the Beijing area at the end of April and September. Figure 15 (B) shows the AOD time series for Seoul (Korea), which is located downwind of East Asia and therefore is an important region for studying aerosols and their transport (Kim et al., 2007; Oh et al., 2015). As shown in Figure 15 (B), Seoul National University (Seoul SNU) is located within the Seoul metropolitan area. In 2022, severe pollution events occurred in April and May. The RMSE for POSP AOD is lower than for MODIS DB, and more valid retrievals were obtained. Figure 15 (C) shows the AOD time series for India, which is one of the most severely air-polluted countries in the world, especially in the northern plains (Vellalassery et al., 2021). As illustrated in Figure 15 (C), Amity University is located in the western part of northern India. In 2022, severe pollution events occurred in May and November. POSP successfully retrieved AOD in April, June, and October, while DB failed to provide results during these periods. This advantage may be attributed to the updated aerosol model for India. Across all stations, the RMSE of POSP is comparable to that for DB AOD. At Beijing CAMS, the RMSE of POSP AOD (0.12) was significantly lower than that of MODIS DB AOD (0.20), highlighting the enhanced precision of POSP in this region.

Figure 15: Time series of the POSP (green diamonds), DB (red triangles) and AERONET AOD (blue circle) over (A) Beijing-CAMS, (B) Amity University, and (C) Seoul University, for January – December 2022.

## **6 Conclusions**



This study focuses on the validation of the newly developed POSP AOD product, processed for the year 2022. To this end, data from 276 global AERONET sites and MODIS/Terra AOD products were used. The POSP AOD was evaluated and analysed in different ways: 1) direct validation versus AERONET reference data at seasonal, regional, and site-specific scales; 2) comparison with similar results for MODIS DT and DB AOD; 3) Effect of land cover types on the AOD retrieval results; 4) evaluation of spatial distribution differences. The principal findings are as follows:

1. The validation of POSP AOD shows good consistency with AERONET AOD, with an R of 0.914, and the fraction within the EE of 78.45%. Global site-scale validation results show that POSP AOD is more consistent with AERONET AOD in high AOD regions than in low AOD regions. The bias is positive in Europe and negative in Asia. The fraction within the GCOS requirements is smaller in high aerosol loading regions than in low aerosol loading regions.

- 2. The accuracy of the POSP AOD varies significantly across different seasons, with the highest accuracy in the DJF (R² of 0.854, RMSE of 0.080) and the lowest in the JJA (R² of 0.667, RMSE of 0.083). The accuracy of the POSP AOD is higher over densely vegetated areas than over low-vegetated areas, with croplands achieving the highest accuracy (R² of 0.859, RMSE of 0.093). Moreover, the error analysis shows that the accuracy of POSP AOD is mainly influenced by surface vegetation cover and observation geometry. As NDVI or scattering angle increases, the uncertainty of POSP AOD decreases. POSP AOD consistently provides results with low bias irrespective of the values of NDVI or scattering angles. For aerosol retrieval over urban areas, the effect of surface anisotropy on retrieval accuracy is non-negligible in regions with high surface reflectance.
- 3. The comparison of MODIS and POSP AOD products shows that POSP AOD is in good agreement with MAIAC AOD over North Africa and the Arabian Peninsula, while it compares better with DB AOD over other regions. Cross-validation shows that the accuracy of the POSP AOD is higher than that of the MODIS AOD. The comparison metrics for DB versus POSP are as follows: R² of 0.853/0.791, RMSE of 0.075/0.090, fraction within EE of 82.51%/77.25% (POSP/DB); and for DT: R² of 0.862/0.770, RMSE of 0.080/0.103, fraction within EE of 80.72%/73.90% (POSP/DT). Comparison over different surface types shows that POSP AOD is more accurate than DB over City, Cropland, and Grassland areas, and better than DT under all surface types.

## Data availability


The POSP/Gaofen-5(02) level-1 data can be downloaded from the website <a href="https://data.cresda.cn/">https://data.cresda.cn/</a>. The AERONET data can be downloaded from the website <a href="https://aeronet.gsfc.nasa.gov/">https://aeronet.gsfc.nasa.gov/</a>. The MODIS/Terra aerosol data can be accessed through the website <a href="https://ladsweb.modaps.eosdis.nasa.gov/">https://ladsweb.modaps.eosdis.nasa.gov/</a>.

## **Auth contributions**

Z.J. conducted the data analysis. Z.L. provided data. Z.J. designed the program code. Z.J. wrote this manuscript. Z.Z., Y.M., Z.S., C.F., GL and Q.Y. reviewed this manuscript. All authors have read and agreed to the published version of the manuscript.

# Competing interests.

The authors declare that they have no conflict of interest.

# Financial support.

This work was supported by the National Natural Science Foundation of China (grant nos.41925019, 42305151) and the National Key R&D Program of China (2023YFB3907405). The participation of Gerrit de Leeuw was supported by the Chinese Academy of Sciences President's International Fellowship Initiative (PIFI) (grant nos. 2025PVA0014).

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
