# Peer review of "Global validation of the Particulate Observing Scanning Polarimeter (POSP) Aerosol Optical Depth products over land"

_EGUsphere, 2025_

## Author Comment (AC3)

Response to RC2:  ['Comment on egusphere-2025-91'](), Anonymous Referee #2, 30 Jun 2025

Review of Manuscript egusphere-2025-91 entitled '**Global validation of the Particulate Observing Scanning Polarimeter (POSP) Aerosol Optical Depth products over land**' by Zhe Ji, Zhengqiang Li, Gerrit de Leeuw, Zihan Zhang, Yan Ma, Zheng Shi, Cheng Fan, and Qian Yao

On behalf of all co-authors, we thank Referee #2 for the insightful and extensive comments which certainly contribute to the substantial improvement of the manuscript (MS). Below we respond to each of the general, major and specific comments which are copied below (in black). In addition to the numbered major and specific comments, we have numbered the general comments as GC1-GC7. After each comment we provide our response, in red, together with changes in the revised MS. Line numbers (indicated by L) mentioned by Referee #2 refer to the original MS as published in the AMT discussion Section and revisions are quoted with line numbers (indicated by LR) referring to the revised MS.

**GC1:** Given that POSP is a new instrument with a novel retrieval algorithm, more detailed information should be provided on the AOD retrieval methodology, particularly regarding the estimation of surface reflectance over land. Since surface reflectance is a critical factor in satellite AOD retrieval, a lack of clarity on how it is treated limits the reader's ability to understand regional differences in retrieval performance. Clear articulation of the algorithm's treatment of land surface properties would help explain spatial variations in validation results.

**Responds to GC1:** Thanks for your suggestion. We have added a description of surface reflectance estimation in Section 2.1, as follows:

"The POSP was launched on board the GF-5(02) satellite in July 2021. It has a field of view of ±50° with a swath width of ~1850 km, and provides global observations in nine spectral bands spanning wavelengths from 380 to 2250 nm (Lei et al., 2023). The local time of the descending node for GF-5(02) is 10:30 a.m. The POSP is equipped with a comprehensive onboard calibration system (the radiometric calibration accuracy is within 5%, and the polarimetric calibration accuracy is within 0.005). Ji et al. (2025) developed an enhanced AOD retrieval algorithm using POSP data. Due to the limited number of observations, POSP faces an ill-posed inversion problem when the directional characteristics of the surface are taken into account. To reduce the discrepancy between the number of observations and the number of retrieval parameters, the following changes have been made to the algorithm presented in Ji et al. (2025). For aerosol parameters, the global aerosol distribution from the MODIS Dark Target algorithm has been used, but aerosol models over northern India and central Africa have been updated to achieve more accurate retrievals. For surface parameters, the bidirectional reflectance distribution function (BRDF) from MODIS (MCD43) was used to account for the directional reflectance characteristics of the surface during the inversion (Schaaf et al., 2002). The MODIS BRDF comprises an isotropic kernel (reflectance from Lambertian surface), a volumetric kernel (reflectance from multiple scattering within vegetation canopies), and a geometric-optical kernel (reflectance from object shadowing). To eliminate the differences in spectral response between POSP and MODIS, spectral reconstruction was performed using the Singular Value Decomposition (SVD) technique. The algorithm only retrieves the isotropic kernel to reduce the number of parameters to be inverted. Therefore, after spectral reconstruction, monthly averaged   Ross-Thick and Li-Sparse kernel parameters were applied to account for the surface directional characteristics. Finally, the new aerosol models and surface directional characteristics were incorporated into the algorithm developed by Ji et al. (2025), and AOD

was successfully retrieved. Ji et al. (2025) also presented the preliminary validation (from November 2021 to April 2022), the results show that the AOD retrievals have high accuracy." (LR 81-100)

**GC2:** The manuscript could benefit from being more concise. Since the primary objective is to validate the POSP AOD product, the content should remain focused on presenting the validation methods, metrics, regional analysis, and interpretation of results, minimizing peripheral discussions.

**Responds to GC2:** We fully agree with your suggestion and have streamlined most of the peripheral discussions. However, following the recommendation of the first reviewer, we have added a discussion on the impact of neglecting surface directional reflectance characteristics on the retrieval over urban areas. In addition, we have also included a comparison of the spatial distribution between POSP AOD and MODIS AOD.

**Responds to GC3::** The current validation extends a previous preliminary comparison (Nov 2021 – Apr 2022) by covering a longer period (Dec 2021 – Nov 2022). However, the manuscript should clearly articulate the novel contributions of this extended study. For instance, does the longer time series reveal seasonal biases? Are regional patterns more robustly confirmed or refined? Clarifying what new insights are gained will better justify the value of this work.

Thanks for your suggestion. We have revised the manuscript and reorganized the highlights of this study, leading to the following conclusions:

This study is dedicated to the following two objectives: 1) To ensure the reliability of POSP AOD products and explore the potential factors influencing their performance; 2) To provide a valuable reference for the enhancement of these products in future iterations.

Firstly, the validation of the POSP AOD against AERONET site data is performed. Then, we obtained the retrieval accuracy of POSP AOD for one year (2022) and the accuracy metrics across different global regions.

"The validation of POSP AOD shows good consistency with AERONET AOD, with an R of 0.914, and the fraction within the EE of 78.45%. Global site-scale validation results show that POSP AOD is more consistent with AERONET AOD in high AOD regions than in low AOD regions. The bias is positive in Europe and negative in Asia. The fraction within the GCOS requirements is smaller in high aerosol loading regions than in low aerosol loading regions." (LR 458-461)

Secondly, we explored the potential factors influencing their performance and specifically discussed the impact of ignoring surface directional reflectance characteristics on the retrieval in urban areas.

"The accuracy of the POSP AOD varies significantly across different seasons, with the highest accuracy in the DJF ($R^2$ of 0.854, RMSE of 0.080) and the lowest in the JJA ($R^2$ of 0.667, RMSE of 0.083). The accuracy of the POSP AOD is higher over densely vegetated areas than over low-vegetated areas, with croplands achieving the highest accuracy ($R^2$ of 0.859, RMSE of 0.093). Moreover, the error analysis shows that the accuracy of POSP AOD is mainly influenced by surface vegetation cover and observation geometry. As NDVI or scattering angle increases, the uncertainty of POSP AOD decreases. POSP AOD consistently provides results with low bias irrespective of the values of NDVI or scattering angles. For aerosol retrieval over urban areas, the effect of surface anisotropy on retrieval accuracy is non-negligible in regions with high surface reflectance." (LR 462-469)

Finally, we analyzed the spatial reliability of POSP AOD by comparing the differences between the POSP AOD and MODIS AOD products.

"The comparison of MODIS and POSP AOD products shows that POSP AOD is in good agreement with MAIAC AOD over North Africa and the Arabian Peninsula, while it compares better with DB AOD over other regions. Cross-validation shows that the accuracy of the POSP AOD is higher than that of the MODIS AOD. The comparison metrics for DB versus POSP are as follows: R² of 0.853/0.791, RMSE of 0.075/0.090, fraction within EE of 82.51%/77.25% (POSP/DB); and for DT: R² of 0.862/0.770, RMSE of 0.080/0.103, fraction within EE of 80.72%/73.90% (POSP/DT). Comparison over different surface types shows that POSP AOD is more accurate than DB over City, Cropland, and Grassland areas, and better than DT under all surface types." (LR 470-476)

**GC4:** Cloud screening is especially crucial for POSP given its spatial resolution of 6.4 km. However, the current manuscript lacks sufficient details on the cloud masking procedures employed. Please describe how cloud contamination is identified and removed from the observations, and discuss the potential impact of residual cloud effects on the validation results.

**Responds to GC4:** Thank you very much for your suggestion. We indeed overlooked the description related to cloud masking. In response, we have added a description of **Data Preprocessing** in Section 3.3 as follows:

"As an optical sensor, POSP observations are inherently susceptible to cloud interference. To mitigate cloud contamination, it is essential to filter out cloud-affected pixels before retrieval. Given the single-angle observation method of POSP, this study adopts cloud detection strategies from MODIS, which have been extensively validated (Frey et al., 2008). Specifically, two methods are employed: the apparent reflectance threshold method and the apparent reflectance spatial variation detection method (Martins et al., 2002). The former effectively identifies optically thick clouds with high reflectance or substantial water vapor content, while the latter is particularly useful for detecting cloud edges, shadows, thin clouds, and dispersed cloud formations.

The land surface exhibits low reflectance in the blue band, whereas clouds have high reflectance. Therefore, a pixel is identified as a cloud when its reflectance at the 443 nm band exceeds a certain threshold. The 1380 nm band lies within a strong water vapor absorption region, where the reflectances from land surfaces and low clouds are generally low. As a result, only high clouds, mostly above the heights where atmospheric water vapor is located, are visible in this band. Pixels with high reflectance at 1380 nm are therefore typically classified as high clouds. Furthermore, cloud edges typically exhibit high spatial variability due to mixed pixels and partial cloud coverage. The spatial variation characteristics of the 443 nm and 1380 nm bands can effectively identify cloud-edge pixels. The combination of their spatial differences helps reduce misclassification at cloud boundaries and improves the accuracy of cloud detection.

Surface conditions such as snow and water also affect the inversion. Since the retrieval algorithm is explicitly designed for clear-sky over non-ice land surfaces, pixels over water, ice, and snow must be excluded. The detection of water and snow pixels is achieved using the Normalized Difference Water Index (NDWI) and the Normalized Difference Snow/Ice Index (NDSI), respectively, with specific identification thresholds presented in Table 1.

$$NDWI = \frac{\rho_{670} - \rho_{865}}{\rho_{670} + \rho_{865}} \tag{9}$$

$$NDSI = \frac{\rho_{670} - \rho_{2250}}{\rho_{670} + \rho_{2250}} \tag{10}$$

While the aforementioned cloud detection strategy provides a foundation for minimizing cloud contamination, potential for further improvement remains. Given the relatively coarse spatial resolution of POSP (6.4 km) and its limited spectral coverage, certain pixels that contain residual clouds may remain undetected. The simulation analysis by Kassianov and Ovtchinnikov (2008) pointed out that multiple scattering of clouds can lead to overestimated AOD retrievals when the residual clouds are not fully screened. Sogacheva et al. (2017) further removed the cloud-contaminated pixels using a cloud post-processing scheme. To enhance cloud-mask accuracy, a dedicated cloud detection algorithm for POSP is still needed. We aim to further enhance the cloud detection algorithm in future work.

Table 1 Summary of screening thresholds.

| Items | Purpose |
|---|---|
| $\rho_{443} < 0.02 \ or \ \rho_{443} > 0.4$ | Cloud |
| $\sigma_{443} > 0.038$ | Cloud |
| $\rho_{1380} > 0.02 \ and \ Height < 1500$ | Cloud |
| $\sigma_{1380} > 0.005$ | Cloud |
| $NDWI > 0$ | Water |
| $NDSI > 0.4$ | Snow/Ice |

" (LR 185-213)

Meanwhile, we have also added a discussion on the potential impact of residual cloud effects on the validation results, as follows:

"Given the relatively coarse spatial resolution of POSP (6.4 km) and its limited spectral coverage, certain pixels that contain residual clouds may remain undetected. The simulation analysis by Kassianov and Ovtchinnikov (2008) pointed out that multiple scattering of clouds can lead to overestimated AOD retrievals when the residual clouds are not fully screened. Sogacheva et al. (2017) further removed the cloud-contaminated pixels using a cloud post-processing scheme. To enhance cloud-mask accuracy, a dedicated cloud detection algorithm for POSP is still needed. We aim to further enhance the cloud detection algorithm in future work." (LR 206-211)

**GC5:** The reference to Che (2015) is cited in the manuscript but not listed in the References section. Please ensure this citation is properly included and formatted.

**Responds to GC5:** Thank you for pointing this out, and we apologize for the confusion caused by our oversight. We have corrected all the reference formats.

**GC6:** The citation "Liangfu et al. (2021)" appears to be incorrect. It should be corrected to "Chen et al. (2021)" as per standard citation format.

**Responds to GC6:** Thank you very much for pointing this out. We have corrected it accordingly.

**GC7:** L85-90, Relying on the high-..., it should be polished.

**Responds to GC7:** We have revised it and removed the inappropriate parts, as follows:

"Ji et al. (2025) also presented the preliminary validation (from November 2021 to April 2022), the results show that the AOD retrievals have high accuracy." (LR 99-100)

**Citation**: https://doi.org/10.5194/egusphere-2025-91-RC2

**References**:

Frey, R. A., Ackerman, S. A., Liu, Y., Strabala, K. I., Zhang, H., Key, J. R., and Wang, X.: Cloud detection with MODIS. Part I: Improvements in the MODIS cloud mask for collection 5, Journal of Atmospheric and Oceanic Technology, 25, 1057–1072, https://doi.org/10.1175/2008JTECHA1052.1, 2008.

Ji, Z., Ma, Y., de Leeuw, G., Shi, Z., and Li, Z.: An enhanced aerosol optical depth retrieval algorithm for Particulate Observing Scanning Polarimeter (POSP) data over land, IEEE Trans. Geosci. Remote Sensing, 63, 1–18, https://doi.org/10.1109/TGRS.2024.3514170, 2025.

Kassianov, E. I. and Ovtchinnikov, M.: On reflectance ratios and aerosol optical depth retrieval in the presence of cumulus clouds, Geophysical Research Letters, 35, https://doi.org/10.1029/2008GL033231, 2008.

Lei, X., Liu, Z., Tao, F., Dong, H., Hou, W., Xiang, G., Qie, L., Meng, B., Li, C., and Chen, F.: Data Comparison and Cross-Calibration between Level 1 Products of DPC and POSP Onboard the Chinese GaoFen-5 (02) Satellite, Remote Sensing, 15, 1933, 2023.

Martins, J. V., Tanré, D., Remer, L., Kaufman, Y., Mattoo, S., and Levy, R.: MODIS cloud screening for remote sensing of aerosols over oceans using spatial variability, Geophysical Research Letters, 29, MOD4-1, 2002.

Schaaf, C. B., Gao, F., Strahler, A. H., Lucht, W., Li, X., Tsang, T., Strugnell, N. C., Zhang, X., Jin, Y., Muller, J.-P., and others: First operational BRDF, albedo nadir reflectance products from MODIS, Remote sensing of Environment, 83, 135–148, 2002.

---

## Author Comment (AC4)

Response to RC1: 'Comment on egusphere-2025-91 ' , Anonymous Referee #1, 11 Mar 2025

Review of Manuscript egusphere-2025-91 entitled '**Global validation of the Particulate Observing Scanning Polarimeter (POSP) Aerosol Optical Depth products over land**' by Zhe Ji, Zhengqiang Li, Gerrit de Leeuw, Zihan Zhang, Yan Ma, Zheng Shi, Cheng Fan, and Qian Yao

On behalf of all co-authors, we thank Referee #1 for the insightful and extensive comments which certainly contribute to the substantial improvement of the manuscript (MS). Below we respond to each of the general, major and specific comments which are copied below (in black). In addition to the numbered major and specific comments, we have numbered the general comments as GC1-GC5. After each comment we provide our response, in red, together with changes in the revised MS. Line numbers (indicated by L) mentioned by Referee #1 refer to the original MS as published in the AMT discussion Section and revisions are quoted with line numbers (indicated by LR) referring to the revised MS.

**GC1:** The specific data quality control procedures for POSP (e.g., cloud detection, outlier removal) remin unclear in the manuscript, potentially affecting result reproducibility. It is recommended to supplement detailed descriptions of POSP data preprocessing steps (e.g., cloud masking, pixel screening criteria) and clarify their impacts on the matching strategy.

**Response to GC1**: Thank you for pointing this out. We overlooked the description of the data preprocessing step, and we appreciate your reminder, which is very helpful in improving the quality of the manuscript. Since the algorithm proposed in this study is specifically designed for cloud-free land pixels, we removed land pixels that might contain clouds or ice/snow before retrieval. When retrieval pixels contain potential cloud contamination, the results tend to be significantly overestimated. A strict cloud detection process can effectively mitigate this issue. Additionally, to ensure the reliability of the validation, we adopted the following matching strategies:

"In this study, considering the 6.4 km spatial resolution of the POSP, the following strategies to match POSP and AERONET AOD data have been devised to ensure reliable AOD validation results while accounting for spatial consistency: satellite data are averaged over a window of 3 × 3 pixels centred on the AERONET site, and ground-based observations are averaged over 30 minutes before and after the time of the satellite overpass. To mitigate the uncertainty associated with averaging data, a minimum of two or more ground-based observations are required in the temporal matchup window, and the spatial-temporal matchup window must encompass more than three valid satellite pixels (Chu et al., 2002)." (LR 145-151)

A detailed description of the preprocessing has been added to the Methods section.

"As an optical sensor, POSP observations are inherently susceptible to cloud interference. To mitigate cloud contamination, it is essential to filter out cloud-affected pixels before retrieval. Given the single-angle observation method of POSP, this study adopts cloud detection strategies from MODIS, which have been extensively validated (Frey et al., 2008). Specifically, two methods are employed: the apparent reflectance threshold method and the apparent reflectance spatial variation detection method (Martins et al., 2002). The former effectively identifies optically thick clouds with high reflectance or substantial water vapor content, while the latter is particularly useful for detecting cloud edges, shadows, thin clouds, and dispersed cloud formations.

The land surface exhibits low reflectance in the blue band, whereas clouds have high reflectance.

Therefore, a pixel is identified as a cloud when its reflectance at the 443 nm band exceeds a certain threshold. The 1380 nm band lies within a strong water vapor absorption region, where the reflectances from land surfaces and low clouds are generally low. As a result, only high clouds, mostly above the heights where atmospheric water vapor is located, are visible in this band. Pixels with high reflectance at 1380 nm are therefore typically classified as high clouds. Furthermore, cloud edges typically exhibit high spatial variability due to mixed pixels and partial cloud coverage. The spatial variation characteristics of the 443 nm and 1380 nm bands can effectively identify cloud-edge pixels. The combination of their spatial differences helps reduce misclassification at cloud boundaries and improves the accuracy of cloud detection.

Surface conditions such as snow and water also affect the inversion. Since the retrieval algorithm is explicitly designed for clear-sky over non-ice land surfaces, pixels over water, ice, and snow must be excluded. The detection of water and snow pixels is achieved using the Normalized Difference Water Index (NDWI) and the Normalized Difference Snow/Ice Index (NDSI), respectively, with specific identification thresholds presented in Table 1.

$$NDWI = \frac{\rho_{670} - \rho_{865}}{\rho_{670} + \rho_{865}} \tag{9}$$

$$NDSI = \frac{\rho_{670} - \rho_{2250}}{\rho_{670} + \rho_{2250}} \tag{10}$$

While the aforementioned cloud detection strategy provides a foundation for minimizing cloud contamination, potential for further improvement remains. Given the relatively coarse spatial resolution of POSP (6.4 km) and its limited spectral coverage, certain pixels that contain residual clouds may remain undetected. The simulation analysis by Kassianov and Ovtchinnikov (2008) pointed out that multiple scattering of clouds can lead to overestimated AOD retrievals when the residual clouds are not fully screened. Sogacheva et al. (2017) further removed the cloud-contaminated pixels using a cloud post-processing scheme. To enhance cloud-mask accuracy, a dedicated cloud detection algorithm for POSP is still needed. We aim to further enhance the cloud detection algorithm in future work.

Table 1 Summary of screening thresholds.

| Items | Purpose |
|---|---|
| $\rho_{443} < 0.02 \; or \; \rho_{443} > 0.4$ | Cloud |
| $\sigma_{443} > 0.038$ | Cloud |
| $\rho_{1380} > 0.02 \; and \; Height < 1500$ | Cloud |
| $\sigma_{1380} > 0.005$ | Cloud |
| $NDWI > 0$ | Water |
| $NDSI > 0.4$ | Snow/Ice |

” (LR 185-213)

**GC2:.** Line24, Lines 178-182: The significant underestimation in high-AOD regions (e.g., North Africa) is attributed to "aerosol model errors" without specific analysis of discrepancies between model assumptions and actual aerosol characteristics. Further investigation into aerosol model classification and its impact on retrieval errors is suggested.

**Response to GC2**: We sincerely apologize for any misunderstanding caused by our oversight. First, Fig. 14 in the manuscript shows that:

"POSP AOD is slightly lower than MODIS DB AOD over North Africa and the Arabian Peninsula, while it is much closer to MODIS MAIAC AOD in these regions. Overall, POSP AOD shows similar features as MAIAC AOD over North Africa and the Arabian Peninsula, while it is more consistent with DB AOD over other regions. Furthermore, compared to MODIS DB, the spatial differences between POSP AOD and MODIS DT are smaller. In general, the results indicate a high degree of agreement between POSP and MODIS AOD, with differences predominantly within the (-0.2, 0.2) range." (LR 424-428)

In recent years, the global ground-based observation network has expanded significantly, improving coverage in many areas. However, ground-based observations remain sparse in remote and inaccessible regions. Given the current distribution of ground-based observation sites, it remains challenging to determine which aerosol product achieves the highest accuracy globally compared to others.

[Figure]

**Figure 14: Maps of the seasonally averaged AOD derived from POSP, MODIS MAIAC, MODIS DB and MODIS DT, for the winter (DJF: December–January–February), spring (MAM: March–April-May,) summer (JJA: June–July–August), and autumn (SON: September–October–November).**

This study performs retrievals based on a fixed aerosol model, which may lead to significant discrepancies between the assumed and actual aerosol models. Li et al. (2018) have quantitatively described the impact of aerosol model error on retrieval accuracy through simulation experiments. They applied the optimal estimation theory and calculated the degree of freedom for signal (DFS) available for aerosol retrieval parameters to quantify their information content (Frankenberg et al., 2012; Hasekamp and Landgraf, 2005). This method has been widely used to assess the theoretical retrieval capability of sensors.

Here, the aerosol model errors correspond with the combination of 6 predefined aerosol parameters: $\{r_{eff}^f,$ $v_{eff}^f, r_{eff}^c, v_{eff}^c, m_i^f, m_i^c\}$, which are all assumed to change from 5% to 100% by a step of 5% with the constant measurement error, as well as the constant a priori errors of $m_i^f$ and $m_i^c$. $r_{eff}^f$ and $r_{eff}^c$ represent the effective radius of fine- and coarse- mode aerosol, respectively. $v_{eff}^f$ and $v_{eff}^c$ represent the effective variance of fine- and coarse-mode aerosol, respectively. $m_i^f$ and $m_i^c$ represent the refractive index of fine- and coarse-mode aerosol, respectively. It is evident that as the aerosol model error increases, DFS decreases linearly, indicating that the retrieval uncertainty correspondingly increases.

[Figure]

**Fig. R1. Same as Fig. 11 but as a function of the aerosol model errors from 5% to 100% by a step of 5% with AOD=0.6.**

We have added descriptions in the relevant sections regarding the impact of aerosol model error on retrieval accuracy.

"The probability density functions of differences (POSP-AERONET) are presented in Figure 2 (B). The results show that the POSP algorithm underestimates the AOD as aerosol loading increases. For low AOD (AOD < 0.2), POSP's bias is 0.01. For moderate AOD (0.2 ≤ AOD ≤ 0.7), POSP's bias increases to -0.03, and for high AOD (AOD > 0.7), POSP's bias further increases to -0.04. These biases may be attributed to the increasing aerosol model error. As AOD increases, the impact of discrepancies between the assumed aerosol model and the actual aerosol model is amplified, leading to an increase in retrieval uncertainty (Hou et al., 2018; Li et al., 2018)." (LR 218-223)

"On the other hand, in heavily polluted regions such as northern India, central and western Africa, and

central South America, POSP AOD shows high consistency with AERONET AOD, although the GCOS fraction is lower. This is because a fixed aerosol model is used to improve the stability of the inversion, which, however, may not accurately represent the actual aerosol types. Such discrepancies introduce greater uncertainties in the retrieval as the aerosol loading increases. Thus, using a fixed aerosol model inevitably affects the retrieval accuracy (Levy et al., 2013). This is one of the inherent challenges of aerosol retrieval using single-angle observations, and we aim to address this issue in future algorithm improvements." (LR 283-288)

**GC3:** Lines 302-304: The explanation for lower AOD accuracy in urban areas remains overly generalized ("complex surface and diverse pollution components"), lacking quantitative analysis (e.g., interference from urban surface reflectance anisotropy). Enhanced discussion on separating urban surface reflectance from aerosol signals is recommended.

**Response to GC3**: Thank you very much for pointing this out. Your comment is extremely valuable for improving the quality of our manuscript. We acknowledge that our analysis lacked a detailed discussion on the impact of urban surface reflectance anisotropy on aerosol retrievals. Therefore, we have now included a comprehensive discussion on this aspect, as detailed below.

[revised manuscript text omitted]

" (LR 329-383)

[Figure]

**Figure S4: The boxplot of the differences for apparent reflectance between the results calculated based on the Lambertian forward radiative transfer model (TOAL) and the non-Lambertian forward radiative transfer model (TOA). The upper panel shows the differences as a function of AOD, while the lower panel presents the differences as a function of the scattering angle.**

[Figure]

**Figure S5: AOD bias as a function of aerosol loading for different urban surface types.**

**GC4:** Line 246:"Other LC types which are not shown in Fig.4 are presented in Fig. S1." Figs. S1-S9 need to be found in the supplementary document. It is recommended to describe clearly in the manuscript.

**Response to GC4**: Thank you for these comments. We have revised the relevant sections to make the explanation as clear as possible.

**GC5:** Some grammatical inconsistencies exist. Comprehensive language polishing is advised to ensure proper tense usage and grammatical consistency throughout the manuscript.

**Response to GC5**: Thank you for these comments. We have substantially revised the MS. We have invited Professor Gerrit de Leeuw, an expert in the field of aerosol remote sensing, to revise our manuscript for grammatical errors and further improve the logic and organization. Furthermore, the manuscript has been carefully read and where necessary, unclear text has been re-formulated.

**Citation**: https://doi.org/10.5194/egusphere-2025-91-RC1